# Critical Appraisal of Systematic Reviews Assessing Gut Microbiota and Effect of Probiotic Supplementation in Children with ASD—An Umbrella Review

**DOI:** 10.3390/microorganisms13030545

**Published:** 2025-02-27

**Authors:** Sachin Agrawal, Chandra Rath, Shripada Rao, Andrew Whitehouse, Sanjay Patole

**Affiliations:** 1Neonatal Directorate, KEM Hospital for Women, Perth, WA 6008, Australia; sachin.agrawal@health.wa.gov.au (S.A.); chandra.rath@health.wa.gov.au (C.R.); 2Perth Children’s Hospital, Perth, WA 6009, Australia; shripada.rao@health.wa.gov.au; 3School of Medicine, University of Western Australia, Perth, WA 6009, Australia; 4Telethon Kids Institute, University of Western Australia, Perth, WA 6009, Australia; andrew.whitehouse@telethonkids.org.au

**Keywords:** autism, probiotics, gut microbiota

## Abstract

Given the significance of gut microbiota in autism spectrum disorder (ASD), we aimed to assess the quality of systematic reviews (SRs) of studies assessing gut microbiota and effects of probiotic supplementation in children with ASD. PubMed, EMBASE, PsycINFO, Medline, and Cochrane databases were searched from inception to November 2024. We included SRs of randomised or non-randomized studies reporting on gut microbiota or effects of probiotics in children with ASD. A total of 48 SRs (probiotics: 21, gut microbiota: 27) were included. The median (IQR) number of studies and participants was 7 (5) and 328 (362), respectively, for SRs of probiotic intervention studies and 18 (18) and 1083 (1201), respectively, for SRs of gut microbiota studies in children with ASD. The quality of included SRs was low (probiotics: 12, gut microbiota: 14) to critically low (probiotics: 9, gut microbiota: 13) due to lack of reporting of critical items including prior registration, deviation from protocol, and risk of bias assessment of included studies. Assuring robust methodology and reporting of future studies is important for generating robust evidence in this field.

## 1. Introduction

Autism spectrum disorder (ASD) is a lifelong neurodevelopmental disorder characterized by difficulties in social communication and interaction as well as the presence of restricted and repetitive behavior [1]. The global prevalence of ASD has increased markedly over the past four decades, with the most recent estimate reporting a global median prevalence of 100/10,000 (range: 1.09/10,000 to 436.0/10,000) [2]. In children born preterm, the risk of ASD is much higher and inversely proportional to their gestational age at birth [3]. Frontline therapies for ASD focus on behavioral and developmental approaches, which seek to support children to acquire key cognitive, language, and daily care skills. However, while these therapies are known to be effective to varying degrees in different children, the high co-occurrence of comorbid conditions, such as gut complaints, leads families to frequently try complementary and alternative approaches, such as dietary interventions, which have been found to have limited efficacy in clinical trials [4,5,6].

In recent years, there has been an increasing focus on the gut microbiome in the pathogenesis of autism considering its interactions and effects via the gut–brain axis. Probiotics, with their potential to correct dysbiosis and influence the gut–brain axis, offer a promising strategy to improve behavioral and social symptoms [7]. Given the plethora of randomized and non-randomized studies assessing gut microbiota and effects of probiotic supplementation in children with ASD, it is not surprising that many SRs have been reported in this field. SRs are at the apex of the pyramid of the hierarchy of evidence-based medicine and are considered important for guiding research and clinical practice and making important health policy decisions [8]. ASD is associated with significant health and socioeconomic burden [9,10]. Given the differences in the findings of systematic reviews of studies assessing gut microbiota and effects of probiotics in ASD, a critical assessment of the quality of the systematic reviews in this field is important. We hence aimed to conduct a critical review of systematic reviews of randomized and non-randomized studies assessing gut microbiota and effects of probiotics in children with ASD.

The aim of the current study was to assess the quality of SRs of studies assessing gut microbiome and effects of probiotic supplementation in children with ASD.

## 2. Materials and Methods

### 2.1. Methods and Participants

A systematic review was conducted to assess the quality of SRs of studies assessing gut microbiota and effects of probiotic supplementation in children with ASD. The assessment of the quality of multiple SRs was conducted using an appropriate tool (AMSTAR-2) [11].

The protocol for this SR was registered on 21 November 2024 in the Open Science registry (No: https://doi.org/10.17605/OSF.IO/RB7VW).

### 2.2. Strategy for Literature Search

We conducted a literature search in July 2024 using PubMed, EMBASE, PsycINFO, Medline, and Cochrane databases from inception to July 2024. We used the following search strategy in PubMed, and similar search strategies were used in other databases: (probiotics) OR (lactobacillus)) OR (Bifidobacterium)) OR (Saccharomyces)) OR (gut microbiota)) OR (gut dysbiosis)) OR (gut microbiome)) OR (gut flora)) AND (((autism) OR (ASD)) OR (autistic)) Filters: Meta-Analysis, Systematic Review. Two reviewers (S.A. and CR) independently screened titles and abstracts from retrieved studies and subsequently full-text articles using predesigned criteria for eligibility. We screened the reference list of potentially relevant articles. Any disagreement between the investigators was resolved by consensus. There was no deviation in the protocol.

### 2.3. Eligibility Criteria: The Criteria for Including SRs in This Review 

(1) SR (with or without meta-analysis) of studies (randomized controlled trials (RCT) or non-randomized (e.g., cohort, case–control) studies of intervention (NRSI) reporting on gut microbiota in children with ASD.

(2) SRs (with or without meta-analysis) of studies (RCTs or NRSIs) reporting the effects of probiotic supplementation (e.g., gut microbiota, behavior symptoms) in children with ASD. 

(3) Full articles available.

Exclusion criteria: (1) Studies in animal models of ASD. (2) Studies involving intervention other than probiotics. (3) Studies where data could not be extracted.

### 2.4. Data Extraction

Two reviewers independently extracted data from each study in a prespecified data collection sheet. The following information was extracted: author name, year of publication, country of origin, type of review with meta-analysis or without, probiotic strain, dosage and schedule of probiotics, number of studies included, total number of participants, age range of participants, aims/objectives, and conclusions. If any study assessed various interventions or outcomes for multiple conditions, data relevant to probiotics or autism were entered where these data was unavailable, and then the combined data were recorded. Data were also added for the studies that used other interventions combined with probiotics.

### 2.5. Assessment of Risk of Bias (RoB)

Two independent authors assessed the methodological quality of included studies using the AMSTAR-2 tool. Discrepancies were resolved by group discussion with the third author. We recorded the answers to all 16 (critical: 9, non-critical: 7) domains. All answers were categorized as “yes” or “no” or “partial yes” or “no meta-analyses”. The studies were graded according to the AMSTAR-2 tool rules [11]. AMSTAR-2 is a critical appraisal tool for SRs of randomized or non-randomized studies of healthcare interventions. It is not intended to generate an overall score and includes a total of 16 items [11].

### 2.6. Domains and Rating

The AMSTAR-2 tool has seven critical domains (items Number 2, 4, 7, 9, 11, 13, 15). The overall rating of the review is provided as high, moderate, low, or critical, depending upon the presence of critical or non-critical weaknesses. 1. High: No or one non-critical weakness: the SR provides an accurate and comprehensive summary of the results of available studies that address the question of interest. 2. Moderate: More than one non-critical weakness: the SR has more than one weakness but no critical flaws. It may provide an accurate summary of the results of the available studies included in the review. 3. Low: One critical flaw with or without non-critical weaknesses: the review has a critical flaw. It may not provide an accurate and comprehensive summary of the available studies that address the question of interest. 4. Critically low: More than one critical flaw with or without non-critical weaknesses: the review has more than one critical flaw and should not be relied on to provide an accurate and comprehensive summary of the available studies.

### 2.7. Interrater Reliability of AMSTAR-2

Cohen’s kappa coefficient (κ scores) is used to measure how well two or more people agree when rating something. Values range from −1 to 1, where 1 indicates complete agreement and 0 signifies no agreement. The inter-rater reliability (κ scores) in the AMSTAR tool was a mean of 0.70 (95% confidence interval [CI]: 0.57, 0.83) (range: 0.38–1.0) across all the domains. Three pairs of raters provided values between 46 and 50 κ scores.

## 3. Results

The PRISMA flow chart for study selection is shown in Figure 1. A total of 1285 SRs were identified on search from databases. Duplicate records (*n* = 585) were removed using endnote. From the remaining 700 citations, 641 were removed after reading the titles and abstract. Forty-eight articles were included in the analyses after reading the full articles. A total of 21 of the included reviews reported outcomes associated with probiotics intervention, and 27 reported outcomes based on microbiota composition. The following articles were excluded [4,12,13,14,15,16,17,18,19,20].

### 3.1. Systematic Reviews: Studies Assessing the Effects of Probiotic Supplementation in Children with ASD

A total of 6 of the included 21 systematic reviews (SRs) focused on RCTs [21,22,23,24,25,26], while 15 included both RCTs and NRSIs [27,28,29,30,31,32,33,34,35,36,37,38,39,40,41]. Eleven SRs examined studies assessed probiotics alongside other interventions for ASD [21,22,23,27,28,29,30,33,36,37,39]. Two SR included studies that reported on psychological conditions in conjunction with ASD [23,40].

Table 1 summarizes the characteristics of the included studies on probiotic interventions. The median (IQR) number of studies included in each SR was 7 (5), with a median (IQR) of 328.5 (362) participants per review. The age of participants across the SRs ranged from 0 to 60 years. Some of the SRs included participants beyond the age of 18 years [26,27,29,40,41].

Out of the 21 included SRs, 15 concluded that there was either no improvement in ASD symptoms or noted a lack of high-quality evidence to draw any definitive conclusions [21,22,23,24,26,28,31,32,33,34,36,37,38,39,41]. A total of 7 of the included 21 SRs conducted a meta-analysis. Six reported no significant improvement in behavioral symptoms associated with ASD [21,22,24,26,34,39]. One of these reviewed a significant improvement in gastrointestinal but not behavioral symptoms associated with ASD [24]. One meta-analysis showed improvement in behavioral symptoms [25]. The methodological quality of all included SRs was rated as low to critically low (Table 2). Individual domain outcomes are reported in Table 3.

### 3.2. Systematic Reviews: Studies Assessing Fecal Microbiota in Children with ASD

Of the 27 SRs (Table 4), 15 that included only NRSIs reported an association of ASD outcomes with fecal microbiota (surrogate of gut microbiota) composition [42,43,44,45,46,47,48,49,50,51,52,53,54,55,56]. A total of 12 SRs included NRSIs as well as RCTs [57,58,59,60,61,62,63,64,65,66,67,68]. The quality of the included SRs assessed by the AMSTAR-2 tool is shown in Table 5. Individual domain outcomes are shown in Table 6. The median (IQR) number of studies included in the SRs was 18 (18), with a median (IQR) of 1083 (1201.75) participants per review. The age of participants in the included reviews ranged from 0 to 60 years. Some of the studies included participants beyond 18 years of age [54,56,69]. A total of 7 out of the 27 studies conducted meta-analysis [43,44,45,51,52,56,58]. Studies assessing fecal microbiota in children with ASD that were included in the SRs reported variable results. Out of 27 SRs, findings of 12 were inconclusive on the role of gut microbiota [42,43,44,55,57,59,60,61,62,64,67,68]. Only three SRs, including one with meta-analysis, suggested that fecal microbiota improved ASD symptoms [47,58,63].

Overall, the quality of SRs of studies reporting on gut microbiota in children with ASD and those assessing the effect of probiotics in this population was downgraded mainly due to the lack of reporting of critical items including prior registration, deviation from protocol, RoB assessment of included studies, and details of excluded studies.

## 4. Discussion

Our review identified 21 SRs (total participants: 5418) reporting on the effect of probiotic supplementation in children with ASD and a further 27 SRs (total participants: 21445) reporting on studies assessing gut microbiota in this population. Overall, the methodological quality of all included SRs, assessed using the AMSTAR-2 tool, was rated as low to critically low. The main reasons for downgrading the quality of included SRs included the lack of reporting of critical items (e.g., prior registration, details of excluded studies, RoB assessment of included studies) and deviation from protocol.

Further key findings emerged from the review of SRs reporting on fecal microbiota of children with ASD. Recent literature suggests that children with ASD have dysbiosis compared to healthy children [69]. The fecal microbiota profile in these children is known to vary from region to region. The common findings include an increase in clostridium species (e.g., *Clostridium bolteae* and *botulinum*), *Bacillus* spp. and *Enterobacteria*, reduction in genera *Prevotella and Roseburia*, and increase in fungal species (e.g., Candida, Saccharomyces). With dysbiosis, there are also changes in amino acids, lipids, and carbohydrate metabolism in gut microbiota. Dysbiosis of gut microbiota related to an overgrowth of pathogenic microbes leads to increased gut permeability, which impairs the integrity of the blood–brain barrier. This allows peripheral neurotoxic proteins or microbial metabolites to enter the brain, leading to neuronal damage or neuroinflammation [69]. A recent SR that included 18 studies (participants: 493 ASD and 404 controls) showed an increase in genera *Bacteroides*, *Parabacteroides*, *Clostridium*, *Fecalibacterium* and *Phascolarctobacterium* and a lower percentage of *Coprococcus* and *Bifidobacterium* in children with ASD. It is postulated that such dysbiosis may explain the behavioral and gastrointestinal symptoms in ASD, mediated via the immune and inflammatory pathways through the gut–brain axis [58]. Levkova et al. reported significantly different bacterial species in children with ASD compared with controls. These included *Bacteroides*, *Bifidobacterium*, *Clostridium*, *Coprococcus*, *Fecalibacterium*, *Lachnospira*, *Prevotella*, *Ruminococcus*, *Streptococcus*, and *Blautia*. The variation in fecal microbiota composition in children with ASD may relate to dietary variations, eating habits, lifestyles, and genetic and environmental influences in this cohort [49,70]. Others report that these variations can also relate to the small sample sizes in such studies apart from the variations in dietary practices, and a high prevalence of functional gastrointestinal symptoms in this population [46].

Similar to the variation in findings observed in the SRs of fecal microbiota, the effects of probiotic supplementation on children with ASD show significant heterogeneity across SRs. The effects of probiotics have been studied in various psychiatric disorders, primarily due to their potential for exerting both local and systemic effects. Locally, probiotics can correct dysbiosis, enhance the gut barrier to prevent or minimize intestinal permeability, reduce inflammation, and modulate neuroendocrine functions. Systemically, they may ameliorate symptoms of ASD by influencing signalling pathways via the gut–brain axis [71]. However, the findings regarding the efficacy of probiotics in children with ASD have been inconsistent. The study by Barba-Vila et al. suggested that probiotics might improve behavioral outcomes and gut microbiota profiles in this population. However, they suggested a note of caution in interpreting their results given the significant methodological limitations across studies. Similarly, a meta-analysis reported by Xiao et al. that included seven clinical trials reported limited efficacy of probiotics in enhancing behavioral symptoms in children with ASD. The authors cautioned against overinterpreting these findings considering the heterogeneity in probiotic strains, duration of supplementation, and other methodological issues in the included studies [34].

A recent meta-analysis by Soleimanpour et al. (eight RCTs, 318 patients with ASD, age: 1.5 to 20 years) showed that the probiotic group had significantly better behavioral symptoms compared to the control group [pooled standardized mean difference (SMD): −0.38 (95% CI: 0.58 to −0.18), *p* < 0.01]. Subgroup analyses involving studies conducted in the European region, those with an intervention period > 3 months, and those focusing on participants under and >10 years of age showed significant benefits. Furthermore, multi-strain [SMD: −0.53 (95%CI: 0.85 to −0.22)] as well as single-strain [SMD: −0.28 (95%CI: 0.54 to −0.02)] probiotics showed significant improvement in behavioral symptoms. The low probability of publication bias supported the validity of the core findings [25].

Understanding the development and evolution of the AMSTAR tool is important before discussing its limitations. First developed by Shea and colleagues in 2007 [72], the AMSTAART tool was created for comprehensive assessment of the methodological quality of SRs of RCTs. The tool, which included 11 items in a questionnaire with options to answer “yes”, “no”, “can’t answer”, or “not applicable”, was not meant for quantitative scoring [72]. Despite the evidence supporting its reliability and validity [73], AMSTAR was criticized for various reasons. These included whether it truly assesses the methodological quality of a SR, its relative lack of adequate guidance and detailed description of publication bias, and difficulty in interpreting some items [74,75]. Introduced in 2010, the revised AMSTAR (R-AMSTAR) used a scoring approach based on domain weight to quantify the quality of a SR [76]. Popovich et al. compared AMSTAR and R-AMSTAR tools for assessment of randomly selected 60 SRs (Cochrane: 30, non-Cochrane: 30) in the field of assisted reproduction. The reviews were graded and ranked based on results converted into percentage scores. The percentage scores showed wider variation and achieved higher inter-rater reliability by AMSTAR vs. R-AMSTAR assessment. The average rating was consistent between the tools (R-AMSTAR: 88.3% vs. AMSTAR: 83.6%) for Cochrane reviews but inconsistent for non-Cochrane reviews (R-AMSTAR: 63.9% vs. AMSTAR: 38.5%). Overall, the Cochrane reviews changed an average of 4.2 places compared to 2.9 for non-Cochrane in the rankings generated between the two tools. The authors concluded that R-AMSTAR provided more excellent guidance in the assessing domains and produced quantitative results. However, AMSTAR was much easier for consistency in application than R-AMSTAR. They recommended incorporating their findings in AMSTAR and additional guidance for its application to improve its reliability and usefulness [77]. AMSTAR-2, the current version of the tool, was developed after extensive revision of the original tool. It includes ten of the original items and six additional items for appraisal of RCTs and non-RCTs. The overall rating is based on weaknesses in critical domains and provides comprehensive guidance for using the tool [11].

The limitations of AMSTAR-2 have been reported by various investigators. Li et al. reported on an AMSTAR 2-based quality assessment of SRs with meta-analysis related to heart failure. A total of 79 of the 81 included SRs were “critically low-quality”. The remaining two were of “low quality.” These findings were attributed to insufficient a priori protocols (compliance rate: 5%), complete list of exclusions with justification (5%), risk of bias assessment (69%), meta-analysis methodology (78%), and investigation of publication bias (60%). The authors concluded that the low rating for these potential high-quality SRs questions the discrimination capacity of AMSTAR 2. They emphasized the need to identify insufficiency areas in the AMSTAR-2 tool and justify or modify its rating rules [78]. Puljak et al. assessed the applicability of AMSTAR-2 to a total of 80 SRs of non-intervention studies in a meta-research study. They used SRs (20 each) for the following four types: diagnostic test accuracy reviews, etiology and/or risk reviews, prevalence and/or incidence reviews, and prognostic reviews. Three authors applied AMSTAR-2 independently to each of the included SRs. They then assessed the applicability of each item to that SRs type and “any SR” type. Their results unanimously showed that 7 of 16 AMSTAR-2 items were applicable for all 4 specific SR types and any SRs (Items 2, 5, 6, 7, 10, 14 and 16), but 8 of 16 items were for any SR type. These items could cover generic SR methods that do not depend on a specific SR type. They concluded that AMSTAR-2 is only partially applicable for non-intervention SRs and emphasized the need to adapt or extend AMSTAR-2 for SR of non-intervention studies [79].

The findings of our umbrella review show the difficulties in deriving robust conclusions from available systematic reviews for guiding research and clinical practice in the field. Current evidence does not help in guiding selection of participants (age at diagnosis, severity of ASD, baseline gut microbiota) or the intervention (e.g., probiotic strain/s, dose, duration). The only finding that is common to all studies in this field is that children with ASD have dysbiosis and probiotic supplementation has the potential to modulate the gut microbiota for host benefits. Interpreting the significance of our results in the context of evidence-based medicine is essential. Well-designed and adequately powered RCTs are considered as the gold standard of clinical research to assess what works and what does not [80]. SRs of RCTs, or non-RCTs as the second-best choice under some circumstances, are at the apex of the pyramid of the hierarchy in evidence-based medicine. Experts advocate that every clinical trial should begin or at least end with a SR [81]. The significance of the quality of SRs cannot be overemphasized for guiding research and clinical practice and making informed health policy decisions. Given the enormous health burden of the underlying health issue and the potential benefits of the proposed intervention, it is important that the design, conduct, and reporting of RCTs and non-RCTs of probiotics to improve the outcomes in young children with ASD needs to be robust. This ensures that the SRs of RCTs and non-RCTs in this field is of high quality. Both the limitations of quality assessment tools such as AMSTAR, as discussed above, and the responsibilities of the investigators in assuring robust design, conduct, and reporting of their RCTs or non-RCTs are critical issues. Conducting adequately powered, well-designed large RCTs to detect small but clinically meaningful effects to improve the outcomes of the participants is challenging, mainly when the participants include community-based children with ASD. Hence, it will not be surprising if the trend towards conducting small studies of probiotics for children with ASD continues. Conducting SRs of small studies will continue to be the accepted approach to generate evidence with more power and precision. However, if information on critical domains of assessment tools such as AMSTAR continues to be missing for various reasons, the quality of SRs of such studies will continue to be low. The current SR adds meaningful data to guide further research in the field of gut microbiota in children with ASD and the potential of probiotics as an intervention for the condition.

## 5. Conclusions

The quality of SRs evaluating gut microbiota and effects of probiotics in children (and in few SRs, adults) with ASD is low to critically low. Robust methodology for designing, conducting, and reporting future clinical studies and their SRs is critical for generating robust evidence to guide research and clinical practice in this field.

## Figures and Tables

**Figure 1 microorganisms-13-00545-f001:**
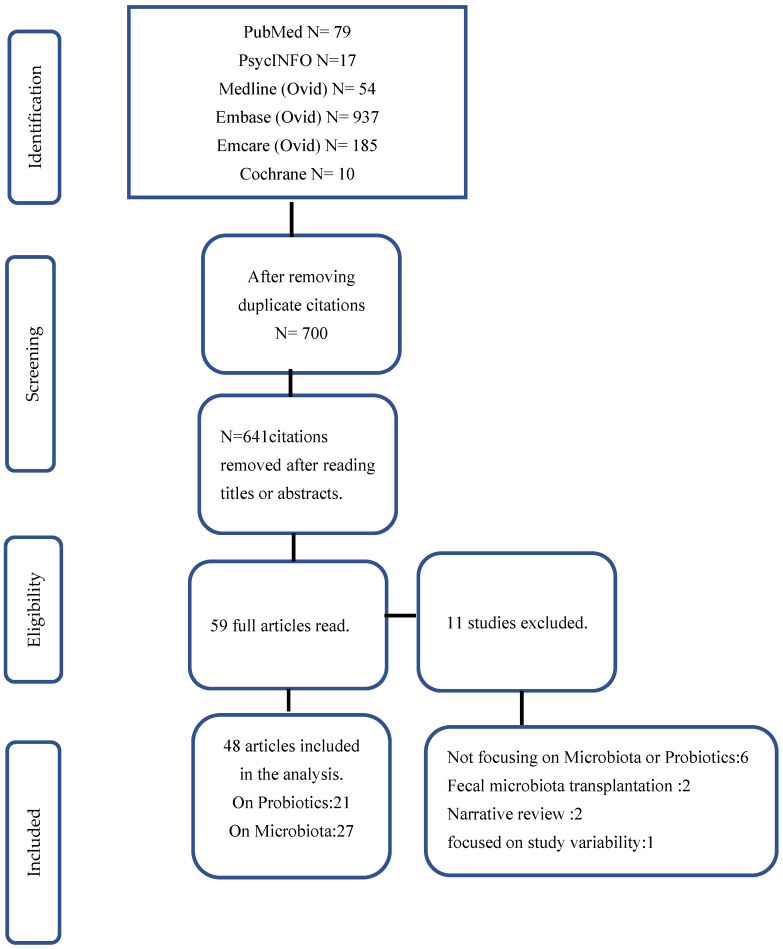
Flow chart of study selection.

**Table 1 microorganisms-13-00545-t001:** Characteristics of included systematic reviews of studies assessing effects of probiotics in children with autism.

s.no.	Study/Country	Year	Design (Type of Studies Included)	Number of Trials (Sample Size)	Study Population	Objectives/Aim	Study Results/Conclusion
1.	Amadi et al./Nigeria [27]	2022	SR of dietary interventions (RCTs and observational studies)	8	2–16-year-olds	Dietary interventions in autism to evaluate their therapeutic efficacy and beneficial effects	Probiotics improved behavioral and GI symptoms as well as restored gut microbiota equilibrium. Prebiotics decreased levels of inflammatory cytokines, improved behavioral and GI symptoms, and improved gut microbiota
2.	Diaz et al./Chile [28]	2022	SR of nutritional intervention of clinical studies	2 (79)	<19-year-olds	To evaluate the effectiveness and the mechanisms involved in nutritional interventions in the mitigation of behavioral symptoms	The five types of nutritional interventions evaluated show varied evidence that does not allow for defining the degree of effectiveness between one or the other in terms of behavioral improvements
3.	Davies et al./US [29]	2021	SR of probiotics and prebiotics (RCT’s, observational open label trials)	7	Children and adolescents	To systematically analyze and discuss the interplay between probiotics and prebiotics to guide further investigation into this field	Probiotic supplementation tended to mitigate some of the behavioral manifestations of ASD, with less of a discernible trend on the microbiome level
4.	Yang et al./China [30]	2020	SR of gut microbial based treatment (RCT and quasi-experiment studies)	11	1–16-year-oldss	To better understand the gut microbiome as a potential therapy for ASD and its role relative to ASD and to qualitatively elucidate microbial-based interventions for ASD	Review provided suggestive evidence regarding the potential beneficial role of gut microbiota in improving behavioral and GI symptoms among ASD patients. These gut microbial-based interventions (i.e., prebiotic, probiotic, vitamin A supplementation, antibiotic, and FMT) provided an insight that gut microbiome may be useful as a potential treatment for ASD patients
5.	Patel/USA [31]	2022	SR of probiotics (SR, RCT’s and observational studies)	9 (710)	0–18-year-olds	Provide a clearer and broader picture of probiotics’ effectiveness and safety profile and also demonstrate different types and combinations of probiotic strains used in several studies, which provide specifications of the most recent data	The use of probiotics in children with ASD might be beneficial for both GI and psychological symptoms, especially for certain specific probiotic strains. The combined use of prebiotics and probiotics seems to be beneficial
6	Alvares/Brazil [32]	2021	SR of probiotics (RCT’s)	2 (134)	Pediatric population	To provide an updated review to clarify the effect of the use of probiotics when compared to placebo in the behavioral aspect and the GIT of pediatric patients with ASD	The approach with probiotics showed low efficacy in improving behavioral symptoms, with some favorable outcomes in patients with GI complaints, which could justify its use in complementary therapies
7	Xiang Ng/China [33]	2019	SR of the role of prebiotics and probiotics (RCT’s, prospective open label studies)	6(495)	2–16-year-olds	Examine the clinical role of prebiotics and probiotics in the management of GI and core ASD symptoms	Despite promising preclinical findings, prebiotics and probiotics have demonstrated an overall limited efficacy in the management of GI or behavioral symptoms in children with ASD
8.	Song/China [21]	2022	SR and MA of probiotics and prebiotics (RCT’s)	3 (144) (74 probiotics and 70 placebo)	4–11-year-olds	Only included clinical controlled trials to explore whether probiotics and prebiotics can improve the overall severity of ASD symptoms in children and the severity of GI problems	Since there are relatively few clinical controlled trials that can be included, the results of this study still need to be further verified in the clinic
9	Siafis/Germany [22]	2022	SR and network MA of pharmacological and dietary–supplements (RCT’s)	5 (121)	Children and adolescents	To better inform clinical practice and identify medications potentially efficacious for ASD, combined evidence from a pharmacological and dietary supplement ASD trial in a network meta-analysis	A large part of the evidence consisted of small RCTs (median 40 participants) with a short duration (median 2 weeks) and limited generalizability
10	Xiao he/China [34]	2023	SR and MA of probiotics (RCT and cross over trials)	10 (460) 7 in meta-analysis	<18-year-olds	It is necessary to expand the scope of the literature search to include more studies on probiotic treatment for children with ASD and conduct a SR and MA to provide new ideas for the probiotic treatment of ASD	Studies provided limited evidence for the efficacy of probiotics due to their small sample sizes, shorter intervention duration, different probiotics used, different scales used, and poor research quality
11	Barbosa/Portugal [23]	2019	SR of Prebiotics and probiotics focuses on psychiatric disorders (RCT’s)	2 (109)	-	To answer the followng question: “What real clinical evidence exists to support the use of probiotics and prebiotics in humans with psychiatric disorders?”	Although recent findings in specific psychiatric disorders are encouraging, the use of prebiotics and probiotics in clinical practice still lack sufficiently robust evidence
12	Patusco/US [35]	2018	Narrative review of role of probiotics in GIT function (RCT and Non-RCTs)	5 (117)	≤18-year-olds	To evaluate the current state of the evidence and provide clinicians with a clearer picture of whether probiotics may benefit their clients with ASD who are experiencing gastrointestinal symptoms and can guide them in addressing professional and parental queries on this practice	There is promising evidence to suggest that probiotic therapy may improve gastrointestinal dysfunction, beneficially alter fecal microbiota, and reduce the severity of ASD symptoms in children with ASD
13	Tan/Canada [36]	2021	SR of Probiotics, prebiotics, Synbiotics, and fecal microbiota Transplantation (RCT’s and open label trial)	8(375)	<18-year-olds	To provide an overview and critically evaluate the current evidence on the efficacy and safety of probiotic, prebiotic, synbiotic, and fecal microbiota transplantation therapies for core and co-occurring behavioral symptoms in individuals with ASD	Current evidence suggests beneficial effects of these modalities in ASD are limited and inconclusive
14	Prosperi/Italy [37]	2022	Intervention on microbiota SR (RCT and open label trial)	13 (689)	<18-year-olds	Offers practitioners an overview of the potential therapeutic options to modify dysbiosis, GI symptoms, and ASD severity by modulating the microbiota–gut–brain axis in ASD, taking into consideration limits and benefits from current findings	Considering the variability of the treatments, the samples size, the duration of treatment, and the tools used to evaluate the outcome, these results are still partial and do not allow us to establish a conclusive beneficial effect of probiotics and other interventions regarding the symptoms of ASD
15	Barba-Vila/Spain [38]	2024	SR of clinical studies of probiotics (RCT’s and open label trials, observational studies)	10	Humans	(1) To give a comprehensive overview of the existing clinical knowledge on the use of probiotics in the treatment of ASD symptoms (behavioral and GI); (2) To assess the near future of this field, propose improvements for novel studies, and make suggestions on how these improvements should be addressed	Although ongoing studies have improved designs, the available knowledge does not permit solid conclusions to be made regarding the efficacy of probiotics in ameliorating the symptoms (psychiatric and/or GI) associated with ASD
16	Rahim/Iraq [39]	2023	MA and umbrella review includes (RCT, cross-over trials and SR)	15	1–60-year-olds	To collect evidence on the efficacy of probiotic, prebiotic, and synbiotic therapy plans. It can aid in formulating well-informed guidelines and procedures for implementing these therapies within the framework of ASD care	Findings revealed that those there was no significant effect of such therapy on autism-related behavioral symptoms. It has a significant effect on the brain connectivity through frontopolar power in beta and gamma bands mediated by chemicals and cytokines, such as TNF-α. The psych biotics showed no serious side effects
17	Liu J/US [41]	2019	SR of Probiotics therapy for treating behavioral and GIT symptoms (RCT and observational studies)	5 (327)	<18-year-olds	To comprehensively assess the efficacy of probiotics in behavioral symptoms of ASD	There is low-quality evidence regarding the potential beneficial role of microbiome dysbiosis in improving behavioral and GI symptoms among ASD patients
18	Zeng/China [24]	2024	MA(RCT’s)	6 (302)	<18-year-olds	To analyze the outcome of probiotics in the treatment of ASD children	Probiotics treatment could improve gastrointestinal symptoms, but there was no significant improvement in ASD
19	Soleimanpour/Iran [25]	2024	SR and MA (RCT’s)	8(396)	1.5–20-year-olds	To examine the effectiveness of probiotic supplementation in alleviating behavioral symptoms in individuals with ASD	Significant improvement in ASD behavioral symptoms through probiotic supplementation
20	Skowron/Switzerland [40]	2022	SR (RCT’s, observational, real-world experience)	5(330)	<25-year-olds	Current knowledge on the influence of psychobiotics on the gut–brain axis on selected disease	Intestinal microbiotas play a relevant role in disorders of the nervous system. The microbiota–gut–brain axis may represent a new target in the prevention and treatment of neuropsychiatric disorders
21	Kotowska/Poland [26]	2024	SR and MA (RCTs)	12(630)	20-year-olds	To systematically assess the evidence on the effects of probiotics on core autism symptoms in children with ASD	Available data do not provide high-quality evidence supporting the use of probiotics for ASD symptoms in children

Gastrointestinal tract (GIT), systematic review (SR) and meta-analysis (MA), randomized control trial (RCT), autism spectrum disorder (ASD), fecal microbiota transplantation (FMT), tumor necrosis factor- alpha (TNF-α).

**Table 2 microorganisms-13-00545-t002:** Risk of bias assessment in studies assessing effects of probiotic supplementation.

s.no.	Study ID	Components of PICO 1	Prior Methodology Established and Deviation * 2	Selection of Study Design Inclusion 3	Comprehensive Literature Search Strategy * 4	At 2 Reviewers for Eligibility 5	At Least 2 Extractors for Data 6	List of Excluded Studies * 7	Details of Included Study 8	ROB of Each Study * 9	Source of Funding 10	Appropriate Statistical Methods * 11	Impact of Individual ROB on Metanalysis 12	Individual ROB * 13	Discuss Heterogeneity 14	Publication of Bias Impact * 15	Conflict of Interest Funding 16	Overall Confidence
1	Amadi 2022 [27]	Yes	Partial yes	Yes	Partial yes	No	No	No	Partial yes	Partial yes	No	No meta-analysis	No meta-analysis	Yes	Yes	No meta-analysis	No	Low
2	Diaz 2022 [28]	Yes	No	Yes	Partial yes	Yes	No	No	No	No	No	No meta-analysis	No meta-analysis	No	Yes	No meta-analysis	No	Critically low
3	Davies 2021 [29]	Yes	Partial yes	Yes	Partial yes	Yes	No	No	Partial Yes	Yes	No	No meta-analysis	No meta-analysis	Yes	Yes	No meta-analysis	Yes	Low
4	Yang 2020 [30]	Yes	Partial yes	Yes	Partial yes	Yes	Yes	No	Partial Yes	Partial yes	No	No meta-analysis	No meta-analysis	No	Yes	No meta-analysis	Yes	Critically low
5	Patel 2022 [31]	Yes	Partial yes	Yes	Partial yes	No	No	No	No	No	No	No meta-analysis	No meta-analysis	No	No	No meta-analysis	Yes	Critically low
6	Alvares 2021 [32]	Yes	Partial yes	Yes	Partial yes	No	No	No	Partial Yes	Partial yes	No	No meta-analysis	No meta-analysis	No	Yes	No meta-analysis	Yes	Critically low
7	Xiang Ng 2019 [33]	Yes	No	Yes	Partial yes	Yes	No	No	Partial Yes	No	No	No meta-analysis	No meta-analysis	No	No	No meta-analysis	Yes	Critically low
8	Song 2022 [21]	Yes	Yes	Yes	Partial yes	Yes	Yes	No	Partial yes	Yes	No	Yes	Yes	Yes	Yes	No	Yes	Critically low
9	Siafis 2022 [22]	Yes	Yes	Yes	Partial yes	Yes	Yes	No	Yes	Yes	Yes	Yes	Yes	Yes	Yes	No	Yes	Critically low
10	Xiao he 2023 [34]	Yes	Yes	Yes	Partial yes	Yes	Yes	No	Partial Yes	Yes	No	Yes	Yes	Yes	Yes	Yes	Yes	Low
11	Barbosa 2019 [23]	Yes	No	Yes	No	No	No	No	Partial Yes	Partial yes	No	No meta-analysis	No meta-analysis	Yes	Yes	No meta-analysis	Yes	Critically low
12	Tan 2021 [36]	Yes	Partial yes	Yes	Partial yes	Yes	Yes	No	Partial Yes	Yes	No	No meta-analysis	No meta-analysis	Yes	Yes	No meta-analysis	Yes	Low
13	Prosperi 2022 [37]	Yes	Partial yes	Yes	Partial yes	Yes	Yes	No	Partial yes	Yes	No	No meta-analysis	No meta-analysis	Yes	Yes	No meta-analysis	Yes	Low
14	Patusco 2018 [35]	Yes	Partial yes	Yes	Partial yes	No	No	No	Partial Yes	Yes	No	No meta-analysis	No meta-analysis	Yes	Yes	No meta-analysis	Yes	Critically low
15	Barba-Vila 2023 [38]	Yes	Partial yes	Yes	Partial yes	Yes	Yes	No	Partial Yes	Yes	No	No meta-analysis	No meta-analysis	Yes	Yes	No meta-analysis	Yes	Low
16	Rahim 2023 [39]	Yes	Partial Yes	Yes	Partial yes	Yes	Yes	No	Partial yes	Yes	No	Yes	Yes	Yes	Yes	Yes	Yes	Low
17	Lin J 2019 [41]	Yes	Partial yes	Yes	Partial yes	Yes	Yes	No	Yes	Yes	No	No meta-analysis	No meta-analysis	Yes	Yes	No meta-analysis	Yes	Low
18	Zeng 2024 [24]	Yes	Partial Yes	Yes	Partial yes	Yes	Yes	No	Partial yes	Yes	No	Yes	Yes	Yes	Yes	Yes	Yes	Low
19	Soleimanpour2024 [25]	Yes	Partial Yes	Yes	Partial Yes	Yes	No	No	Partial Yes	Yes	No	Yes	Yes	Yes	Yes	Yes	Yes	Low
20	Skowron 2022 [40]	Yes	Partial Yes	Yes	Partial Yes	Yes	No	No	Partial Yes	Partial yes	No	No meta-analysis	No meta-analysis	Yes	Yes	No meta-analysis	Yes	Low
21	Kotowska2024 [26]	Yes	Partial yes	Yes	Partial yes	Yes	Yes	No	Partial yes	Yes	Yes	Yes	Yes	Yes	Yes	Yes	Yes	Low

* Critical domains: 2, 4, 7, 9, 11, 13, 15.

**Table 3 microorganisms-13-00545-t003:** AMSTAR-2 domains in studies of probiotic supplementation.

Domain Number	* C/NC	Domain Content	Yes/Partial Yes (%)	No (%)	No ^#^ MA (%)
1	NC	Components of PICO	100	0	
2	C	Prior methodology established and deviation	85.7	14.3	
3	NC	Selection of study design inclusion	100	0	
4	C	Comprehensive literature search	94.5	5.5	
5	NC	At least 2 reviewers for eligibility	76.2	23.8	
6	NC	At least 2 extractors for data	52.4	47.6	
7	C	List of excluded studies	0	100	
8	NC	Details of included studies	90.5	9.5	
9	C	ROB of each study	85.8	14.2	
10	NC	Source of funding	9.5	94.5	
11	C	Appropriate statistical methods	33.4	0	66.6
12	NC	Impact of individual ROB on MA	33.4	0	66.6
13	C	Individual ROB	76.1	23.9	
14	NC	Discuss heterogeneity	90.5	9.5	
15	C	Publication of bias impact	23.9	9.5	66.6
16	NC	Conflict of interest funding	90.5	9.5	

Abbreviations: * critical: C, non-critical: NC, ^#^ MA: meta-analysis.

**Table 4 microorganisms-13-00545-t004:** Characteristics of included systematic reviews of studies assessing gut microbiota in children with autism.

s.no.	Study/Country	Year	Study Design and Type of Studies Included	Number of Trials (Sample Size)	Study Population	Study Question/Aim	Meta Analysis Results/Conclusion
1	Bezawada/UK [57]	2020	SR (RCT s and observational Studies)	28 (1702)	2–18-year-olds	Systematically review the existing literature to evaluate variations in the gut microbiota and understand its significance in ASD	The gut microbiota is altered in ASD, although further exploration is needed on whether this is a cause or an effect of the condition
2	Cao et al./China [42]	2013	SR (observational studies and open clinical trial)	15 (805) (437 ASD and 368 controls)	2–18-year-olds	To evaluate and summarize findings from studies on the characteristics of the GI microbiome in children with ASD	Further studies are needed
3	Lucía Iglesias-Vázquez/Spain 2020 [58]	2020	SR and MA (RCTs and observational studies)	18 (897) (493 ASD and 404 control)	2–18-year-olds	Update current findings about the composition of gut microbiota in children and adolescents with ASD	This meta-analysis suggests that there is a dysbiosis in ASD children which may influence the development and severity of ASD symptomatology
4	Pedro Andreo-Martínez/Spain [56]	2021	SR and MA (observational studies)	18 (998) (642 ASD and 356 control)		To carry out a meta-analysis on the studies that analyzed GM in children with ASD	Our results showed a lower relative abundance of Streptococcus (SMD+ = −0.999; 95% CI −1.549, −0.449) and Bifidobacterium genera (SMD+ = −0.513; 95% CI −0.953, −0.073) in children with ASD
5	Mingyu Xu/China [43]	2019	SR and MA (observational studies)	9 (421) (254 ASD and 167 control)	6–11-year-olds	To better understand the effect of gut microbes on ASD, a meta-analysis was carried out to assess the differences in microbial populations between patients with ASD and age-matched control	An association between ASD and alteration of microbiota composition warrants additional prospective cohort studies to evaluate the association of bacterial changes with ASD symptoms, which would provide further evidence for the precise microbiological treatment of ASD
6	Ligezka/US [59]	2021	SR of microbiota in neuropsychiatric disorder (RCTs and observational studies)	4	0–18-year-olds	To systematically review the current evidence base regarding human gut microbiota and its role in neuropsychiatric disorders	More studies are needed to determine whether gut dysbiosis leads to the development and/or contributes to the severity of mental disorders or whether gut dysbiosis is a result of other processes that accompany mental disorders
7	Lewandowska-Pietruszka/Poland [60]	2023	SR (SR, RCT, MA, or open label or observational study)	44 (1939) (1123 ASD and 816 controls)	0–18-year-olds	To present the current state of knowledge about the microbiota composition as well as the interventional possibilities that were observed to be effective in single studies in these patients	The altered composition of the gut microbiota in children with ASD and its correlation with GI symptoms and core behavioral characteristics warrants further investigation
8	Martínez-González/Spain [61]	2019	SR (observational studies and RCTs)	16	Children	To compare the GM of people with ASD and GI symptoms with those of healthy controls	It is still too early to draw a conclusion about the gut microbes involved in GI symptoms of ASD
9	Srikantha/Switzerland [62]	2019	SR (SR, RCT, MA, or open label or observational study)	136	-	To collect all peer-reviewed human studies and reviews on the topic connecting gut microbiota with ASD and suggesting metabolites, which could serve as potential biomarkers after further validation	The correlation between changes in distinct bacterial populations and several bacterial metabolites and the behavioral changes related to ASD warrant further investigation
10	West/USA [44]	2022	MA (observational studies)	10 (690)	<18-year-olds	To undersatnd the heterogeneity of the patient groups, leading to the identification of appropriate patient subsets to deliver targeted therapeutics in the future	Gut microbiomes of the ASD population exhibit appreciable heterogeneity, an observation that has been established regarding the clinical manifestations of the disorder
11	Wu/China [45]	2020	MA (observational studies	5 (297) (169 ASD and 128 control)		To verify the claims in previous studies concerning changes in the gut microbiome associated with ASD	Genera Prevotella, Roseburia, Ruminococcus, Megasphaera, and Catenibacterium might be biomarkers of ASD after LEfSe evaluation and Random Forest analysis, respectively
12	Zafar/Pakistan [63]	2021	SR (SR, RCT, MA, or open label or observational study)	8(508) (330 ASD and 178 controls)	-	To determine the association between the gut microbiota and the gut–brain axis in children with autism	The microbiota composition of the gut does affect the manifestations of ASD. The derangement of the gut commensals may lead to mood disorders, depression, and other symptoms in autistic kids
13	Hua ho/Singapore [64]	2020	SR (observational studies and RCTs)	26	<18-year-olds	To take a closer look at whether the plethora of literature published provides consistent evidence on features of gut microbiome alterations associated with ASD and to establish the strength of evidence	These results were inadequate to confirm a global microbiome change in children with ASD and causality could not be inferred to explain the aetiology of the behaviors associated with ASD. Mechanistic studies are needed to elucidate the specific role of the gut microbiome in the pathogenesis of ASD
14	Caputi/US [46]	2024	SR of microbiome in ASD, ADHD and Rett syndrome (observational studies)	18 (2251)	<18-year-olds	To collect SRs of alterations of the intestinal microbiota composition and function in paediatric and adult patients affected by ASD, ADHD, and RETT syndrome	Several discrepancies were found among the included studies, primarily due to sample size, variations in dietary practices, and a high prevalence of a functional gastrointestinal symptom
15	Liu F/China [47]	2019	SR (observational studies)	16 (664) (381 ASD and 283 control)	2–18-year-olds	This SR aimed to explore the current evidence for the alteration of gut microbiota in ASD patients compared with healthy controls using culture-independent techniques	Significant alterations of gut microbiota were demonstrated in ASD patients compared with HCs, which strengthens the evidence that dysbiosis of gut microbiota may correlate with behavioral abnormality in ASD patients. Results of inconsistent changing also exist. Further big-sampled well-designed studies are needed
16	Bonnechere/Belgium [48]	2022	SR of the role of Gut Microbiota in Neuropsychiatric Diseases (observational studies)	23 (1724) (932 and 792 control)	2–18-year-olds	To determine whether gut microbiota are implicated consistently to a specific disorder across studies and whether the associations unique to the disease or whether the associations are also seen in other psychiatric diseases.	This study provides new insights into the complex relationship between the brain and the gut and the implications in neuropsychiatric pathologies. The identification of unique signatures in neuropsychiatric diseases suggests new possibilities in targeted anti- or probiotic treatment
17	Levkova/Bulgaria [49]	2023	Mini review (observational studies)	34	<18-year-olds	To list the bacterial genera for which there is the greatest evidence of a substantial difference between patients with ASD and typically developing children	Demonstrated that the abundance of various bacterial genera within the gut microbiome of children with ASD varies across different studies. Despite significant differences observed, the same genera exhibited decreased levels in some studies while increased levels were reported in other studies
18	Bundagaard-Nielsen/Denmark [50]	2020	Gut microbiota profile in ASD and ADHD-SR (observational studies)	20 (1323) (733 cases and 590 controls)	<18-year-olds	To investigate and describe the current findings relating to altered gut microbiota composition in individuals with ASD and ADHD	Demonstrated that ASD and ADHD cases are associated with a gut microbiota different from controls without neurodevelopmental disorders. However, studies varied widely concerning methodology, resulting in highly heterogeneous gut microbiota compositions between studies
19	Zhang/China [51]	2023	MA (observational studies)	26 (1972) (1021 ASD and 951 controls)	2–17-year-olds	Review and meta-analysis to identify differences in gut microbiota profiles between ASD and TD as well as to provide a theoretical basis for targeted interventions and treatments for ASD through gut microbiota	Results demonstrated relatively up-regulated abundance of Bacteroidetes, Verrucomicrobia, Bacteroides, Clostridium, Dorea and *Sutterella*, and down-regulated abundance of *Proteobacteria*, *Bifidobacterium*, *Coprococcus*, *and Akkermansia* in ASD children, indicating partial agreement in the ASD-associated microbes despite the heterogeneity of ASD
20	Korteniemi/Finland [65]	2023	SR (observational studies and RCTs)	51	2–18-year-olds	To demonstrate interrelations between ASD and the gut microbiota in children based on scientific evidence	These results show that the gut microbiota of children with ASD is altered compared to one of neurotypically developed children
21	Nitschke/Canada [66]	2020	SR (observational, RCTs open label studies)	13	Children	To demonstrate ASD and its association with gut microbiota	Consistent evidence reveals a close association between ASD and GI dysfunction, hence supporting a role of the MGB axis in the pathogenesis of ASD
22	Lin P/Japan [52]	2024	MA (observational studies)	43	<18-year-olds	To comprehensively evaluate a wide range of potential biomarkers associated with ASD	Association between the levels of CRP, GABA, oxytocin, Fe, Zn and the relative abundance of Bifidobacterium, Parabacteroides, Bacteroides, Clostridium, and ASD, suggesting that these indications may be promising biomarkers for ASD
23	Chen/Sweden [53]	2021	SR of gut microbiota in psychiatric disorder (observational studies)	29	No restriction on age	To compare gut microbiota in patients with a psychiatric disorder	Gut bacteria that produce short-chain fatty acids, such as Roseburia and Fecalibacterium, could be less abundant in patients with psychiatric disorders, whereas commensal genera, Bifidobacterium, might be more abundant compared with healthy controls
24	Grau-De Valle/Spain [54]	2023	SR of gut microbiota in psychiatric disorder (observational studies)	4	<18-year-olds	To highlight the relationship between gut microbiota and psychiatric disorders	Reduction in fermentative taxa has been observed in different psychiatric disorders, resulting in a decrease in the production of short-chain fatty acids (SCFAs) and an increase in pro-inflammatory taxa, both of which may be consequences of the exacerbation of these pathologies
25	Jurek/France [55]	2020	SR of dysbiosis in neurodevelopmental disorder (NDD) (Observational studies)	28 (3002)	9-month–27-year-olds	1. To assess whether there is any microbiome modification in participants with NDD compared to controls; 2. Whether one of these features is associated with NDD severity	Comprehensive evidence that “dysbiosis in NDD” is not a well-validated scientific hypothesis
26	Lasheras/Spain [67]	2020	SR (RCTs, case series and case control studies)	9	-	To demonstrate effects of modulation of gut microbiota in ASD patients	Preliminary data show microbiota-based therapies to have a positive effect on ASD patients. However, further well-designed, large-scale randomized controlled trials with standardized protocols are needed to support the effectiveness and safety of these treatments
27	Alamoudi/Australia [68]	2022	SR gut microbiome in autism and preclinical models: (open label, observational studies and RCTs)	13(1169) (623 ASD and control 546)	No age restriction	To demonstrate that further characterization of microbiome profiles across both clinical and animal model datasets is needed to pinpoint how microbial changes impact gut physiology in people with ASD	Studies in both humans and mice highlighted multidirectional changes in abundance (i.e., in some cases abundance increased whereas other reports showed decreases)

Linear discriminant analysis effect size (LEfSe), gut microbiota (GM), autism spectrum disorder (ASD), attention deficit hyperactivity disorder (ADHD), microbiota gut–brain axis (MGB), C-reactive protein (CRP), systematic review (SR), meta-analysis (MA), neurodevelopmental disorder (NDD), gamma-amino butyric acid (GABA).

**Table 5 microorganisms-13-00545-t005:** Risk of bias assessment in studies assessing gut microbiota.

	Study ID	Components of PICO 1	Prior Methodology Established and Deviation * 2	Selection of Study Design Inclusion 3	Comprehensive Literature Search Strategy * 4	At 2 Reviewers for Eligibility 5	At Least 2 Extractors for Data 6	List of Excluded Study * 7	Details of Included Study 8	ROB of Each Study * 9	Source of Funding 10	Appropriate Statistical Methods * 11	Impact of Individual ROB on Metanalysis 12	Individual ROB * 13	Discuss Heterogeneity 14	Publication of Bias Impact * 15	Conflict of Interest Funding 16	Overall Confidence
1	Bezawada 2020 [57]	Yes	Partial yes	Yes	Partial yes	Yes	Yes	No	No	Partial yes	No	No meta-analysis	No meta-analysis	Yes	Yes	No meta-analysis	Yes	Low
2	Cao 2013 [42]	Yes	Partial yes	Yes	Partial yes	Yes	Yes	No	No	Partial yes	No	No meta-analysis	No meta-analysis	Yes	Yes	No meta-analysis	Yes	Low
3	Lucía Iglesias-Vázquez 2020 [58]	Yes	Yes	Yes	Partial Yes	Yes	Yes	No	No	Partial yes	No	Yes	Yes	Yes	Yes	No	Yes	Low
4	Pedro Andreo Martinez 2021 [56]	Yes	Partial yes	Yes	Partial yes	Yes	Yes	No	No	Yes	No	Yes	Yes	Yes	Yes	Yes	No	Low
5	Mingyu Xu 2019 [43]	Yes	Partial yes	Yes	Partial Yes	Yes	Yes	No	Partial yes	No	No	Yes	Yes	Yes	Yes	No	Yes	Critically low
6	Ligezka 2021 [59]	Yes	Partial yes	Yes	Partial Yes	Yes	Yes	No	No	Partial yes	No	No meta-analysis	No meta-analysis	Yes	Yes	No meta-analysis	Yes	Low
7	Lewandowska-Pietruszka 2023 [60]	Yes	Partial yes	Yes	Partial Yes	No	No	No	Partial Yes	Yes	No	No meta-analysis	No meta-analysis	Yes	Yes	No meta-analysis	Yes	Low
8	Martínez-González 2019 [61]	Yes	Partial yes	Yes	Partial Yes	Yes	No	No	No	Partial yes	No	No meta-analysis	No meta-analysis	Yes	Yes	No meta-analysis	Yes	Low
9	Srikantha 2019 [62]	Yes	No	Yes	No	Yes	No	No	No	No	No	No meta-analysis	No meta-analysis	No	No	No meta-analysis	Yes	Critically low
10	West 2022 [44]	No	No	Yes	Partial Yes	No	No	Yes	No	No	No	Yes	Yes	Yes	Yes	No	Yes	Critically low
11	Wu 2021 [45]	Yes	No	Yes	No	No	No	No	No	No	No	Yes	Yes	No	Yes	Yes	Yes	Critically low
12	Zafar 2021 [63]	Yes	No	Yes	No	No	No	No	Partial Yes	No	No	No meta-analysis	No meta-analysis	No	No	No meta-analysis	No	Critically low
13	Hu ho 2020 [64]	Yes	Partial yes	Yes	Partial yes	Yes	No	No	No	No	No	No meta-analysis	No meta-analysis	Yes	Yes	No meta-analysis	Yes	Critically low
14	Caputi 2024 [46]	Yes	Partial yes	Yes	Partial yes	Yes	Yes	No	Partial yes	Partial yes	No	No meta-analysis	No meta-analysis	Yes	Yes	No meta-analysis	Yes	Low
15	Liu F 2019 [47]	Yes	Partial yes	Yes	Partial yes	Yes	Yes	No	No	Yes	No	No meta-analysis	No meta-analysis	Yes	Yes	No meta-analysis	Yes	Low
16	Bonnechere 2022 [48]	Yes	Partial yes	Yes	Partial yes	No	No	No	No	No	No	No meta-analysis	No meta-analysis	No	Yes	No meta-analysis	Yes	Critically low
17	Levkova 2023 [49]	Yes	No	Yes	Partial yes	No	No	No	No	No	No	No meta-analysis	No meta-analysis	No	Yes	No meta-analysis	Yes	Critically low
18	Bundagaard-Nielsen 2020 [50]	Yes	Partial yes	Yes	Partial yes	Yes	No	No	No	Partial yes	No	No meta-analysis	No meta-analysis	Yes	Yes	No meta-analysis	Yes	Low
19	Zhang/2023 [51]	Yes	Yes	Yes	Partial yes	No	No	No	No	Partial yes	No	Yes	Yes	Yes	Yes	No	Yes	Critically low
20	Korteniemi/ 2023 [65]	Yes	No	Yes	No	No	No	No	No	No	No	No meta-analysis	No meta-analysis	No	Yes	No meta-analysis	Yes	Critically low
21	Nitschke/2020 [66]	Yes	No	Yes	Partial yes	No	No	No	Partial Yes	No	No	No meta-analysis	No meta-analysis	Yes	Yes	No meta-analysis	Yes	Critically low
22	Lin P/2024 [52]	Yes	Partial yes	Yes	Partial yes	No	No	No	Partial Yes	Yes	No	Yes	Yes	Yes	Yes	Yes	Yes	Low
23	Chen/2021 [53]	Yes	Partial Yes	Yes	Partial yes	Yes	Yes	No	No	Partial Yes	No	No meta-analysis	No meta-analysis	Yes	Yes	No meta-analysis	Yes	Low
24	Grau-De Valle 2023 [54]	Yes	Partial yes	Yes	Partial yes	No	No	No	Partial yes	Partial Yes	No	No meta-analysis	No meta-analysis	Yes	Yes	No meta-analysis	Yes	Low
25	Jurek 2020 [55]	Yes	Partial yes	Yes	Partial yes	Yes	Yes	No	No	Partial yes	No	No meta-analysis	No meta-analysis	Yes	Yes	No meta-analysis	Yes	Low
26	Lasheras 2020 [67]	Yes	Partial yes	Yes	No	Yes	Yes	No	Partial yes	No	No	No meta-analysis	No meta-analysis	No	Yes	No meta-analysis	Yes	Critically low
27	Alamoudi/ 2022 [68]	Yes	Partial yes	Yes	Partial yes	No	No	No	No	Partial yes	No	No meta-analysis	No meta-analysis	Yes	Yes	No meta-analysis	Yes	Critically low

* Critical domains: 2, 4, 7, 9, 11, 13, 15.

**Table 6 microorganisms-13-00545-t006:** AMSTAR-2 domain outcomes in studies of gut microbiota.

Domain Number	Critical(C) or Non-Critical (NC)	Domain Content	Yes, or Partial Yes (%)	No (%)	No Meta-Analysis (%)
1.	NC	Components of PICO	96.3	3.7	
2.	C	Prior methodology established and deviation	77.8	22.2	
3.	NC	Selection of study design inclusion	100	0	
4.	C	Comprehensive literature search strategy	81.5	18.5	
5.	NC	At least 2 reviewers for eligibility	55.6	44.4	
6.	NC	At least 2 extractors for data	40.8	59.2	
7.	C	List of excluded study	3.7	96.3	
8.	NC	Details of included study	29.6	70.4	
9.	C	ROB of each study	59.2	40.8	
10.	NC	Source of funding	0	100	
11.	C	Appropriate statistical methods	25.9	-	74.1
12.	NC	Impact of individual ROB on metanalysis	25.9		74.1
13.	C	Individual ROB	74.1	25.9	
14.	NC	Discuss heterogeneity	92.6	7.4	
15.	C	Publication of bias impact	7.4	18.5	74.1
16.	NC	Conflict of interest funding	92.6	7.4	

## Data Availability

No new data were created or analyzed in this study. Data sharing is not applicable to this article.

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
