# Peer review of "Critical Appraisal of Systematic Reviews Assessing Gut Microbiota and Effect of Probiotic Supplementation in Children with ASD—An Umbrella Review"

_microorganisms, 2025, doi:10.3390/microorganisms13030545_

Round 1

Reviewer 1 Report

Comments and Suggestions for Authors

Thank you very much for your letter and for the reviewers’ comments concerning our manuscript titled " Critical appraisal of systematic reviews assessing gut microbiota and effect of probiotic supplementation in children with ASD - an umbrella review" (ID: microorganisms-3455890-peer-review-v1). The work aims to address the role of probiotic supplementation in maintaining health in ASD.  

Although the manuscript generally reads well, closer inspection has raised some minor concerns that should be addressed by the authors.

The abstract is too long and needs to be shortened. Specifically, the amount of the results presented in the abstract needs to be summarized more efficiently.

Please include the level of evidence in table 2.

Author Response

  1. Comment: The abstract is too long and needs to be shortened. Specifically, the amount of the results presented in the abstract needs to be summarized more efficiently.

Response: As suggested, we have reduced the word count of the abstract focussing mainly on the results.

  1. Comment: Please include the level of evidence in table 2.

Response: Thanks for the suggestion. To our knowledge there is no validated and widely accepted method for grading the “level of evidence” in an umbrella review. Table 2 and 4 provide the ‘ratings’ of the included systematic reviews given by the AMSTAR tool. We will be happy to add the ‘level of evidence’ based on the reviewer’s advice.

Reviewer 2 Report

Comments and Suggestions for Authors

line 5: and *

line 18: pubmed and medline is same

line 26: add specific number “the quality of included srs was low to critically low due to lack of”

line 29: expand on implications of the findings for future research or and clinical practice.

line 46: is not clear why umbrella review is needed. what are the known methodological limitations of srs in this field, and why is their quality particularly important in asd research?

line 64: the use of amstar-2 is appreciated pls upload them as supplementary.

line 65: prisma is now prisma2020 and it is not specific to umbrella reviews.

line 71: pubmed and medline is same

line 79: report κ scores for inter-rater reliability + include a brief explanation of κ scores and their significance

line 87: english only is a limitation

line 88: justify the exclusion of non-probiotic interventions

line 88: how reviewer disagreements were resolved

the results are presented in a structured manner i commend separating findings on probiotics and gut microbiota it was really easy to track.

heterogeneity is not presented and discussed within srmas.

the discussion does not fully address the implications of the findings for microbiology and clinical practice e.g, how might dysbiosis in asd populations inform future probiotic interventions or gut-brain axis therapies? what specific microbiota-targeted strategies that appear more promising based on the srs reviewed?

the discussion of microbiota findings can be enriched. the authors mention changes in specific bacterial genera (e.g., clostridium, bacteroides) but do not elaborate on their potential mechanisms of action or clinical relevance.

technical writing and formatting tables are too cut maybe improve their format

Author Response

  1. Comment: Line 5: and *

Response: We have now deleted ‘and *’

  1. Comment: Line 18: pubmed and medline is same

Response: We searched both, PubMed and Medline because of the potential advantages. The PubMed search engine allows access to most recent publications which have not been indexed by the Medline team based on the MeSH terms. There is usually a 3-month delay. Direct search of Medline allows using MeSH terms, an advantage as investigators may not be fully aware of different search terms for the same variable.

  1. Comment: Line 26: add specific number “the quality of included srs was low to critically low due to lack of”

Response: We have included the number of SRs with low to critically low ratings for probiotic intervention and in microbiota studies

  1. Comment: Line 29: expand on implications of the findings for future research or and clinical practice.

Response: We have modified the lines (27-29) as follows: “Assuring robust methodology and reporting of future studies is important for generating robust evidence in this field.”

  1. Comment: Line 46: is not clear why umbrella review is needed. what are the known methodological limitations of srs in this field, and why is their quality particularly important in asd research?

Response: We have improvised the line justifying the need for an umbrella review as follows: “ASD is associated with significant health and socioeconomic burden [9,10]. Given the differences in the findings of systematic reviews of studies assessing gut microbiota and effects of probiotics in ASD, a critical assessment of the quality of the systematic reviews in this field is important.” (Lines 53 to 56)

  1. Comment: Line 64: the use of amstar-2 is appreciated pls upload them as supplementary.

Response: We have uploaded the AMSTAR tool as a supplement.

  1. Comment: Line 65: prisma is now prisma 2020 and it is not specific to umbrella reviews.

Response: We agree that PRISMA is not used for ‘Umbrella’ reviews.  Except for the title, the strategy for reporting the results of the systematic search of SRs of interest is identical to that shown by the flow of PRISMA diagram. To avoid confusion, we have deleted the reference to PRISMA, and used the legend as ‘Flow diagram for search strategy and results.’

  1. Comment: Line 71: pubmed and medline is same

Response: Please see our response to comment 2.

  1. Comment: Line 79: report κ scores for inter-rater reliability + include a brief explanation of κ scores and their significance

Response: We have now included this information as follows: “Cohen’s kappa coefficient (κ scores) is used to measure how well two or more people agree when rating something. The values of kappa coefficient range from -1 to 1, where 1 indicates complete agreement and 0 signifies no agreement The mean [95% confidence interval (CI)] inter-rater reliability (κ scores) in AMSTAR tool was 0.70 (CI: 0.57, 0.83, range: 0.38–1.0) across all the domains.” (Lines 120-124)

  1. Comment: Line 87: English only is a limitation

Response: Apologies for the error. We aimed to exclude studies with no access to full text. This criteria did not relate to ‘English’. We have deleted the word ‘English’ from the concerned sentence. Furthermore, unavailability of full text was not the reason for exclusion for the 11 excluded studies.

  1. Comment: Line 88: justify the exclusion of non-probiotic interventions

Response: Non-probiotic interventions were beyond the scope of our review. Furthermore, such interventions have been reported to have limited efficacy in clinical trials (line 43-44).

  1. Comment: Line 88: How reviewer disagreements were resolved

Response: As described in line 77, disagreements were resolved by consensus among the reviewers

  1. Comment: The results are presented in a structured manner i commend separating findings on probiotics and gut microbiota it was easy to track.

Response: Thanks for the comment,

  1. Comment: Heterogeneity is not presented and discussed within srmas.

Response: We have covered the possible reasons for heterogeneity (e.g. variation in probiotic strain, dose, and duration, study design i.e. RCTs vs non-RCTs) in the discussion section (Lines 286- 292,line 304-306)

  1. Comment: The discussion does not fully address the implications of the findings for microbiology and clinical practice e.g, how might dysbiosis in asd populations inform future probiotic interventions or gut-brain axis therapies? what specific microbiota-targeted strategies that appear more promising based on the srs reviewed?

Response: The findings of our umbrella review show the difficulties in deriving robust conclusions from available systematic reviews for guiding research and clinical practice in the field. Current evidence does not help in guiding selection of participants (age at diagnosis, severity of ASD, baseline gut microbiota) or the intervention (e.g. probiotic strain/s, dose, duration). The only finding that is common to all studies in this field is that children with ASD have dysbiosis and probiotic supplementation has the potential to modulate the gut microbiota for host benefits. Only well-designed and definitive RCTs can help in this context.

We have covered these implications of our findings in the revised manuscript. (lines 363-369).

  1. Comment: The discussion of microbiota findings can be enriched. the authors mention changes in specific bacterial genera (e.g., clostridium, Bacteroides) but do not elaborate on their potential mechanisms of action or clinical relevance.

Response: We have added this information (Lines: 274-277) as follows: “Dysbiosis of gut microbiota related to an overgrowth of pathogenic microbes leads to increased gut permeability and impaired integrity of the blood–brain barrier. This allows peripheral neurotoxic proteins or microbial metabolites to enter the brain, leading to neuronal damage or neuroinflammation”.

  1. Comment: Technical writing and formatting tables are too cut maybe improve their format

Response: We have edited the entire manuscript to address these issues.

Reviewer 3 Report

Comments and Suggestions for Authors

Dear authors,

Thank you for your interesting review, I have a few questions and comments:

1.      Line  5  - “Sachin Agrawal 1 , Chandra Rath 1 , Shripada Rao 2,5, Andrew Whitehouse3, Sanjay Patole 4,5 and *” – Have you forgotten to list someone as a co-author?

2.      Table 2 - in my opinion it would be more convenient if you added a numbering to make it easier to correlate the data in Table 2 with the data in Table 1.

3.      Line 262 – “.4. Discussion” – Remove the extra point.

4.      Line 263 – Remove the repetition of the section title.

5.      Line 264 – “Total participants: 5418” – Which participants do you mean?

6.      Line 389 – “The quality of SRs evaluating gut microbiota and effects of probiotics in children…” – Why do the authors only talk about children in their conclusions? The manuscript contains tables with data on the assessment of reviews of studies that included more than children.

Author Response

  1. Comment: Line 5 - “Sachin Agrawal 1 , Chandra Rath 1 , Shripada Rao 2,5, Andrew Whitehouse3, Sanjay Patole 4,5 and *” – Have you forgotten to list someone as a co-author?

Response: We have deleted extra “and”.

  1. Comment: Table 2 - in my opinion it would be more convenient if you added a numbering to make it easier to correlate the data in Table 2 with the data in Table 1.

Response: We have now included these numbers in table for easy referencing.

  1. Comment: Line 262 – “.4. Discussion” – Remove the extra point.

Response: We have now deleted extra ‘.’

  1. Comment: Line 263 – Remove the repetition of the section title.

Response: We have removed the repeated section title.

  1. Comment: Line 264 – “Total participants: 5418” – Which participants do you mean?

Response: These are the total number of participants included in 21 SRs reporting the effect of probiotic supplementation in children with ASD.

  1. Comment: Line 389 – “The quality of SRs evaluating gut microbiota and effects of probiotics in children…” – Why do the authors only talk about children in their conclusions? The manuscript contains tables with data on the assessment of reviews of studies that included more than children.

Response: We have added the number (and corresponding citation number) of SRs that included participants beyond the paediatric age group (lines 187 and 238).  This is now covered briefly in the final ‘Conclusion’.

Round 2

Reviewer 2 Report

Comments and Suggestions for Authors

thanks for addressing my. comments